# Product selectivity in plasmonic photocatalysis for carbon dioxide hydrogenation

Xiao Zhang[1], Xueqian Li[1], Du Zhang[1], Neil Qiang Su[1], Weitao Yang[1], Henry O. Everitt[2,3] & Jie Liu[1]

Photocatalysis has not found widespread industrial adoption, in spite of decades of active research, because the challenges associated with catalyst illumination and turnover outweigh the touted advantages of replacing heat with light. A demonstration that light can control product selectivity in complex chemical reactions could prove to be transformative. Here, we show how the recently demonstrated plasmonic behaviour of rhodium nanoparticles profoundly improves their already excellent catalytic properties by simultaneously reducing the activation energy and selectively producing a desired but kinetically unfavourable product for the important carbon dioxide hydrogenation reaction. Methane is almost exclusively produced when rhodium nanoparticles are mildly illuminated as hot electrons are injected into the anti-bonding orbital of a critical intermediate, while carbon monoxide and methane are equally produced without illumination. The reduced activation energy and super-linear dependence on light intensity cause the unheated photocatalytic methane production rate to exceed the thermocatalytic rate at 350 °C.

[1] Department of Chemistry, Duke University, Durham, North Carolina 27708, USA. [2] Department of Physics, Duke University, Durham, North Carolina 27708, USA. [3] Army Aviation & Missile RD&E Center, Redstone Arsenal, Alabama 35898, USA. Correspondence and requests for materials should be addressed to H.O.E. (email: everitt@phy.duke.edu) or to J.L. (email: j.liu@duke.edu).

The societal need for industrial scale catalysis continues to grow in response to increasing demands for fertilizer, fuels and materials. For heterogeneous catalytic reactions with large activation energies to achieve practical rates, heated catalysts are used, but these demand high energy inputs, shorten catalyst lifetimes through sintering deterioration[1,2], and require product selectivity to mitigate unfavourable side reactions. Rising atmospheric carbon dioxide ($CO_2$) concentrations may be reduced, for example, by reacting ambient $CO_2$ with renewably generated hydrogen ($H_2$)[3,4], but product selectivity is essential to ensure production of hydrocarbons such as methane ($CH_4$) instead of the kinetically preferred product carbon monoxide (CO)[5,6]. Selective $CO_2$ hydrogenation is also an essential purification step for the feedstock used in ammonia synthesis to fix nitrogen for fertilizers[7]. Ideal catalysts simultaneously lower operating temperatures, accelerate reaction rates, and preferentially select products without being consumed or altered. In spite of extensive research on the subject, no photocatalyst has yet achieved this lofty objective. Semiconductor-based photocatalysts offer a promising route to room temperature reactions[8–10], but they exhibit limited selectivity[11] and reaction rates that typically scale only as the square root of the light intensity ($R_{photo} \propto I^{0.5}$)[12,13], making it impractical to increase the reaction rate by increasing light intensity.

Recently, it has been discovered that plasmonic metal nanoparticles are photocatalytically active,[14–29] driving chemical reactions with photo-generated hot carriers and exhibiting a compelling super-linear dependence on light intensity ($R_{photo} \propto I^n$, $n > 1$)[18,28,30]. Plasmonic metal nanoparticles are characterized by strong light absorption through excitations of collective free electron oscillations, called localized surface plasmon resonances (LSPRs) that may be spectrally tuned throughout the visible or ultraviolet by choice of metal, size, shape and host medium. Of particular interest is the decay of LSPRs into hot carriers and their subsequent transfer to adsorbates where they may affect reaction pathways and rates[31–34]. The distribution of photoexcited carriers depends on the local density-of-states in the metal and the associated band structure, the LSPR of the nanostructure and the energy of the photon[35]. By tuning photon and LSPR energies so that hot carriers are injected into specific anti-bonding orbitals of specific reaction intermediates, product selectivity may be achieved[26,36,37].

These early demonstrations of plasmonic photocatalysis either featured intense laser pulses ($\sim kW\,cm^{-2}$) on nanoparticle clusters to generate high concentrations of hot carriers[14,16–18], or they used alloyed or hybrid nanostructures composed of plasmonic (gold, silver, aluminium) and catalytic (platinum, cobalt, palladium) metals[20,21,26–28]. The ideal photocatalytic metal should simultaneously exhibit good plasmonic and catalytic behaviors to increase the rates and selectivity of the reaction[25]. Recently, the size- and shape-dependent plasmonic properties of rhodium (Rh) nanoparticles have been demonstrated at energies tunable throughout the ultraviolet and visible regions[38–43]. Like Au and Pt, Rh is a transition metal without a native oxide coating, and direct bonding between adsorbates and the metal surface greatly facilitates the transfer of hot carriers for plasmonic photocatalysis. Supported Rh nanoparticles and molecular complexes are widely used as catalysts in automotive catalytic converters to reduce nitrogen oxides, as well as in industrial hydrogenation, hydroformylation, and ammonia oxidation reactions[44–46].

Here, we report the discovery that plasmonic Rh nanoparticles are photocatalytic, simultaneously lowering activation energies and exhibiting strong product photo-selectivity, as illustrated through the $CO_2$ hydrogenation reaction. $CO_2$ hydrogenation on

transition metals at atmospheric pressure proceeds through two competing pathways: $CO_2$ methanation ($CO_2 + 4H_2 \rightarrow CH_4 + 2H_2O$) and reverse water gas shift (RWGS, $CO_2 + H_2 \rightarrow CO + H_2O$)[47]. We observe that mild illumination of the Rh nanoparticles not only reduces activation energies for $CO_2$ hydrogenation $\sim 35\%$ below thermal activation energies, it also produces a strong selectivity towards $CH_4$ over CO. Specifically, under illumination from low-intensity ($\sim W\,cm^{-2}$), continuous wave blue or ultraviolet light-emitting diodes (LEDs), the photocatalytic reactions on unheated Rh nanoparticles produce $CH_4$ with selectivity of $> 86\%$ or $> 98\%$, respectively, with a reaction rate twice that of the thermocatalytic reaction rate at 623 K (350 °C). This high selectivity towards $CH_4$ disappears when the Rh nanoparticles are not illuminated, in stark contrast to plasmonic gold (Au) nanoparticles that only catalyse CO production whether illuminated or not. Density functional theory (DFT) calculations indicate the photo-selectivity of the Rh photocatalyst can be attributed to the alignment of the hot electron distribution with the anti-bonding orbital of the critical reaction intermediate, CHO, which activates the $CO_2$ methanation pathway. Our discovery that plasmonic Rh nanoparticles exhibit a photocatalytic activity with strong product photo-selectivity opens an exciting new pathway in the long history of heterogeneous catalysis by offering a compelling advantage of light over heat.

## Results

**Photocatalytic and thermocatalytic reactions.** The Rh photocatalyst was prepared by dispersing 37 nm Rh nanocubes on aluminium oxide ($Al_2O_3$) nanoparticles with a mass loading of 1.02% (Rh/$Al_2O_3$, Fig. 1a)[38]. The synthetic method used (see 'Methods' section) produces cubic nanoparticles whose size and LSPR wavelength can be precisely tuned, and whose sharp corners concentrate light and liberate hot carriers[48]. For these experiments, the 334 nm (3.71 eV) LSPR of the Rh nanocubes in ethanol is broadened and blue-shifted on a porous $Al_2O_3$ support but still overlapped our 365 nm (3.40 eV) ultraviolet light source (Fig. 1b; Supplementary Fig. 1). A blue LED (460 nm, 2.70 eV) was also used to study the influence of excitation wavelength. The band structure of Rh (Supplementary Fig. 2)[49] indicates that the ultraviolet and blue excitations avoid lower energy parasitic interband absorption and generate nearly free hot electrons with energies 2.5 eV and 2.1 eV above the Fermi level[35,41], respectively. The Au photocatalyst was supported on an $Al_2O_3$ support with a mass loading of 1.70% (Au/$Al_2O_3$)[15,50]. The Au nanoparticles have a spherical shape with an average diameter of 2.6 nm (Supplementary Fig. 3) and an LSPR near 517 nm (2.40 eV). A white LED (400–800 nm) was used for the Au photocatalyst, unless otherwise stated. Figure 1b plots the strong absorption of the Rh and Au photocatalysts in the ultraviolet and visible regions, respectively, and their relationship with the emission spectra of the LEDs.

A fixed-bed reaction chamber equipped with a quartz window was used to carry out the photocatalytic reactions with controlled light illumination (Fig. 1e). The photocatalysts were packed with a thickness of $\sim 4\,mm$ ($\sim 15\,mg$) to ensure complete absorption of light. For the heated experiments, the temperature of the powder catalysts was precisely measured with a thermocouple and controlled by resistive heating and cooling water. However, the unheated 'ambient' experiments were initiated at room temperature and used no cooling water, so the temperature was allowed to rise and equilibrate. A mass spectrometer was connected to the chamber outlet for real-time, quantitative analysis of gaseous products. The conversion of $CO_2$ was maintained $< 5\%$ to eliminate reactant-transport limitations and

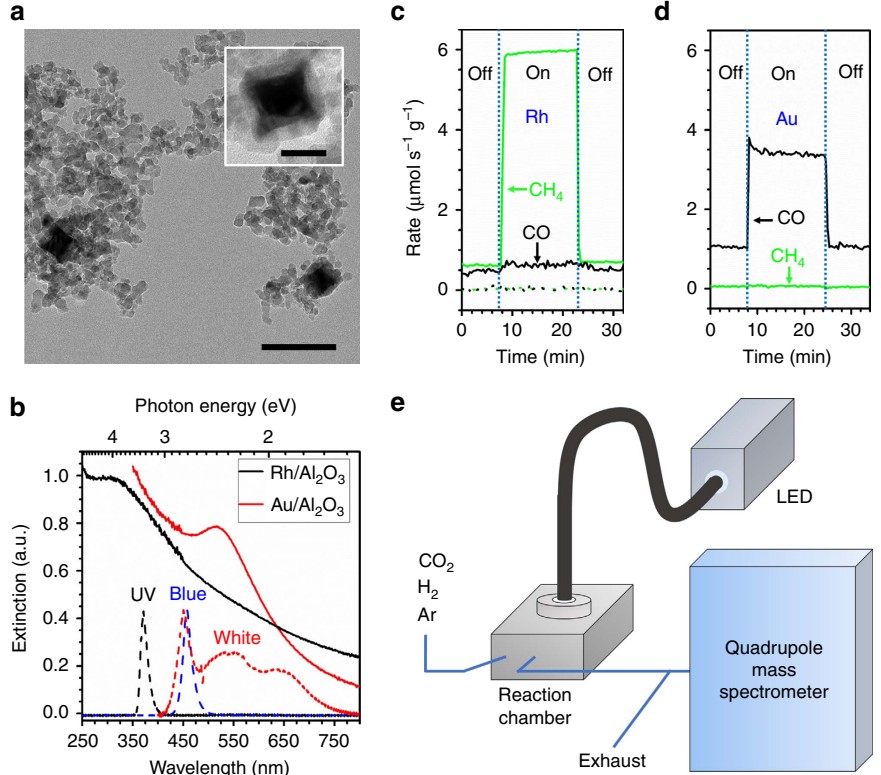

**Figure 1 | CO$_2$ hydrogenation on the rhodium and gold photocatalysts.** (**a**) TEM images of the Rh/Al$_2$O$_3$ photocatalyst. Scale bar, 100 nm (inset: 25 nm). (**b**) Ultraviolet–visible extinction spectra (solid lines) of the Rh/Al$_2$O$_3$ (black) and Au/Al$_2$O$_3$ (red) photocatalysts, measured by diffuse reflectance in an integrating sphere, overlaid with the emission spectra (dotted lines) of the ultraviolet (black), blue (blue) and white (red) LEDs. (**c**) Rates of CH$_4$ (green) and CO (black) production at 623 K on Rh/Al$_2$O$_3$ (solid lines) and Al$_2$O$_3$ (dotted lines) under dark and ultraviolet illumination at 3 W cm$^{-2}$. CH$_4$ production is strongly and selectively enhanced by ultraviolet light on the Rh photocatalyst. Neither CH$_4$ nor CO production was detected on Al$_2$O$_3$. (**d**) Rates of CO (black) and CH$_4$ (green) production at 623 K on Au/Al$_2$O$_3$ under dark and white light illumination at 3 W cm$^{-2}$. A light-enhanced reaction rate is observed, but CO remains the exclusive product under both conditions. (**e**) Schematic of the photocatalytic reaction system, consisting of a stainless steel reaction chamber with a quartz window, LEDs coupled through a light guide, and a mass spectrometer for product analysis.

ensure that the concentrations of products in the effluent represent the reaction rates. In dark conditions, the reaction rates represent the thermocatalytic activities of the catalysts, while under illumination, the overall reaction rates are considered as the sum of thermo- and photocatalytic contributions. Thus, the photoreaction rates are obtained by subtracting the thermal reaction rates (light off) from the overall reaction rates (light on) at the same temperature. All reactions were performed at atmospheric pressure with either an H$_2$-rich 1:5.5 or H$_2$-deficient 1:3.1 mixture of CO$_2$:H$_2$ (as compared with the 1:4 stoichiometry of CO$_2$ methanation) and argon (Ar) as an internal standard.

**Selectivity**. On the Rh photocatalyst (Fig. 1c, solid lines), CH$_4$ and CO were produced at comparable rates at 623 K under dark, H$_2$-rich conditions (for example, 0–8 min). Upon illumination of ultraviolet light (for example, 8–22 min), a seven-fold increase in the CH$_4$ production rate was observed, while only a slight increase in CO production was detected. No other carbon-containing product was observed above the detection limit of the mass spectrometer in our experiments, and the reaction rates responded to light instantly and reversibly. Control experiments using pure Al$_2$O$_3$ nanoparticles (Fig. 1c, dotted lines) and isotopic labelling experiments with deuterium (Supplementary Fig. 4) confirmed that CH$_4$ and CO were produced from the photocatalytic reactions on the Rh nanocubes rather than from contaminants or the Al$_2$O$_3$ support. Comparable photo-enhanced

CO$_2$ hydrogenation was also observed on the Au photocatalyst under white light illumination of similar intensity (Fig. 1d), but with distinctly different product selectivity: CO was the exclusive carbon-containing product on the Au photocatalyst under both dark and light conditions. Even under the same ultraviolet illumination as the Rh photocatalyst, CO was exclusively produced on the Au photocatalyst (Supplementary Fig. 5), indicating that wavelength alone cannot account for the different selectivity.

These results demonstrate that the different selectivity of thermo- and photocatalytic reactions on the Rh and Au nanoparticles is primarily determined by the properties of metals, specifically the differing metal-adsorbate interactions. On the Rh catalysts, previous experimental and theoretical investigations[47,51–58] showed that CO$_2$ first dissociatively adsorbs on the Rh surface to generate adsorbed CO and oxygen (O). The adsorbed CO can either desorb from the surface or be hydrogenated to form CHO. The dissociation of CH–O generates CH, followed by further hydrogenation to form CH$_4$ (see Supplementary Fig. 6 for the detailed mechanism). The desorption of CO from the metal surface was identified as the rate-determining step (RDS) of CO production, and the dissociation of CH–O was the RDS for CH$_4$ production[47]. Thus, competition between CO desorption and C–O bond cleavage in CHO dictates the product selectivity. The O adsorption energy, $E_{ads,O}$, on late transition-metal surfaces is known to be an effective descriptor of the selectivity of CO$_2$ hydrogenation[47]. A high $E_{ads,O}$ partially compensates the energy cost for C–O bond

cleavage in the CHO intermediates and increases the selectivity towards $CH_4$. Although the reaction on the Au catalysts has been reported to involve additional reaction intermediates[22,59,60], the selectivity observed here is consistent with the corresponding $E_{ads,O}$ of Rh (5.22 eV) and Au (3.25 eV)[61]: the Rh catalyst had a slight preference towards $CH_4$ production under dark conditions, whereas the Au catalyst exclusively produced CO.

The selectivity of these reactions is changed when hot carriers are photoexcited in plasmonic nanoparticles. The different selectivity of thermo- and photo-reactions on Rh nanoparticles is depicted in Fig. 2a,b. The dark thermocatalytic reaction exhibited mild selectivity, with a $CH_4$:CO ratio of ~60:40 in the tested range of temperatures and reaction rates. In contrast, under ultraviolet illumination the $CH_4$ production rate was significantly and selectively enhanced. The photoreactions exhibit >95% selectivity towards $CH_4$, and the resulting selectivity towards $CH_4$ from the overall reaction is >90% under 3 W cm$^{-2}$ ultraviolet illumination and $H_2$-rich conditions over the tested temperature range. Experiments under $H_2$-deficient conditions maintained this high selectivity under illumination but exhibited even lower selectivity under dark conditions, confirming that illumination, not heat or excess $H_2$ feedstock, is responsible for the highly selective production of $CH_4$ (Fig. 2b). The Rh nanoparticles and the porous $Al_2O_3$ support are in excellent thermal equilibrium

with each other ($\Delta T \ll 1$ K, Supplementary Note 1 and refs 62,63) because of their physical contact and high thermal conductivities. The rather modest local heating in our experiments and the observed high photo-selectivity towards $CH_4$ over CO indicate that the photo-enhanced reaction rates do not originate from thermal or plasmonic photothermal heating on the Rh nanoparticle surface. Instead, it is the plasmon-generated hot electrons that selectively activate CHO intermediates and accelerate $CH_4$ production while minimally affecting the CO-metal bond for CO production (desorption). This analysis is based on the assumption that thermo- and photo-reactions have the same elementary steps and surface intermediates, a claim supported by a recent kinetic study of RWGS on Au photocatalysts[22]. Although the selectivity is derived almost entirely from light, heat significantly increases the reaction rate.

For the ambient experiments, efficient photocatalytic $CH_4$ production with high selectivity was demonstrated on Rh under ultraviolet illumination at 3 W cm$^{-2}$ and $H_2$-rich conditions with a reaction rate (circled red square in Fig. 2d) comparable to the thermocatalytic reaction rate at 548 K (275 °C). The slightly elevated steady-state temperature, measured to be 328 K ($\Delta T = 29$ K), was caused by photothermal heating of the catalyst bed (separately measured to be 25 K by a non-reactive control experiment with flowing Ar and $H_2$) and the exothermic nature

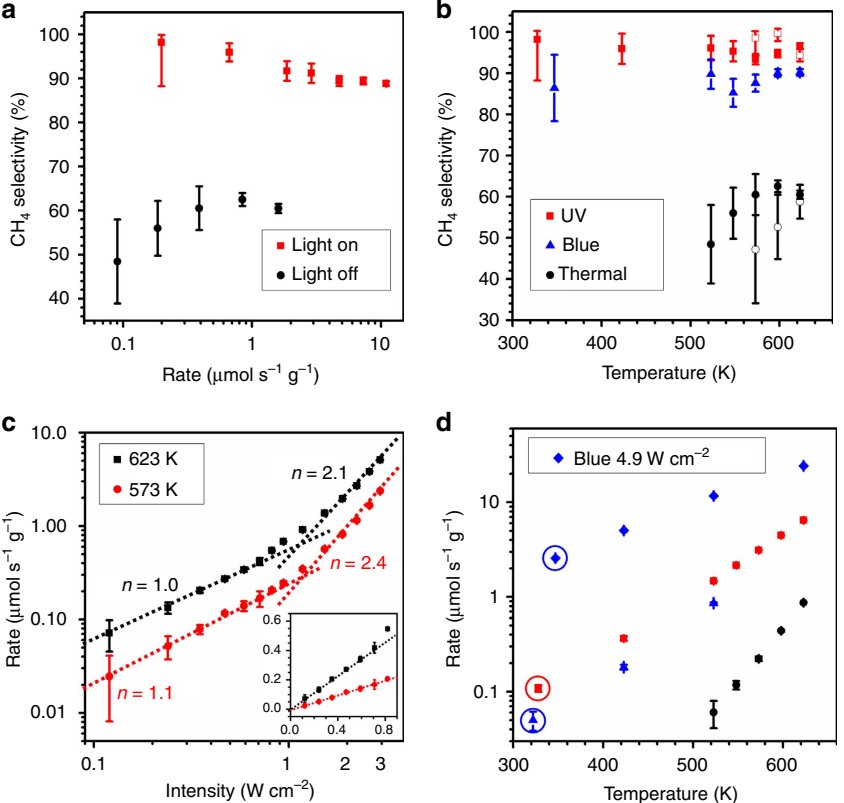

**Figure 2 | Product selectivity and reaction rates on the rhodium photocatalyst.** (**a**) Selectivity towards $CH_4$ as a function of overall reaction rates in dark (black circles) and under ultraviolet light at 3 W cm$^{-2}$ (red squares). (**b**) Selectivity towards $CH_4$ of the thermo- (black circles) and photocatalytic reactions under ultraviolet (365 nm, red squares) and blue (460 nm, blue triangles) illumination as a function of temperature under $H_2$-rich (CO$_2$:$H_2$ = 1:5.5, solid symbols) and $H_2$-deficient (CO$_2$:$H_2$ = 1:3.1, open symbols) conditions. The photoreaction rates are calculated by subtracting the thermocatalytic reaction rates from overall reaction rates at the same temperature. The photoreactions under ultraviolet light show higher selectivity towards $CH_4$ than under blue light, which are both much higher than that of the thermocatalytic reaction. (**c**) Rates of $CH_4$ photo-production as a function of ultraviolet light intensity at 623 (black squares) and 573 K (red circles). The intensity-dependent reaction rates show a linear to super-linear transition with increasing light intensity. The inset shows the intensity-dependent reaction rates in the linear region. (**d**) Overall, $CH_4$ production rates in dark (black circles) and under ultraviolet (red squares, 3 W cm$^{-2}$) and blue (blue triangles, 2.4 W cm$^{-2}$) LEDs with the same photon flux, and with twice the blue photon flux (blue diamonds, 4.9 W cm$^{-2}$). Ultraviolet light is more efficient at enhancing the reaction rates than blue light with the same photon flux. Circled points show the unheated steady-state temperatures and reaction rates. Error bars represent the s.d. of measurements by the mass spectrometer.

of the $CO_2$ methanation reaction (4 K, $\Delta H^0 = -165.01\,kJ\,mol^{-1}$ at 298 K). Likewise, the ambient reaction rate for the highest intensity of our blue LED (4.9 W cm$^{-2}$, $\Delta T = 48$ K, circled blue diamonds in Fig. 2d) was two times higher than that of the thermocatalytic reaction rate at 623 K (350 °C) with a quantum yield, defined as the molar ratio of methane generated to photons delivered (Supplementary Note 2), of 0.82%. It is important to recognize that these high reaction rates with high selectivity were achieved using an efficient, low-intensity LED.

## Discussion

The effects of LED intensity and photon energy on the reaction rates using the Rh photocatalyst were carefully studied by varying the output power and wavelength of the light source. Under ultraviolet illumination near $\sim 1$ W cm$^{-2}$, the photoreaction rate under $H_2$-rich conditions changed from a linear to a super-linear dependence on light intensity ($R_{photo} \propto I^n$, $n = 2.1$ at 623 K and 2.4 at 573 K, Fig. 2c). This super-linear relationship confirms that the photoreactions are mediated by hot electrons[18] and can be attributed to multiple excitations of the vibrational mode(s) of the adsorbed RDS intermediate by hot electrons[30]. In the low-intensity linear region, the slope is significantly higher at 623 K than at 573 K (Fig. 2c, inset) as heat accelerates the photocatalytic rate. Conversely, the photocatalytic reaction rates were greatly enhanced at the highest intensity of the ultraviolet LED (3 W cm$^{-2}$, red squares in Fig. 2d), compared with the thermocatalytic reaction rates at the same temperature. The enhancement factor for $CH_4$

production (EF = $R_{overall}/R_{thermal}$) was 7.41 ± 0.37 at 623 K and increased to 24.4 ± 1.2 at 523 K. The quantum yield for $CH_4$ production was measured to be 3.70% at 623 K.

Under illumination from the blue LED with the same photon flux (2.4 W cm$^{-2}$) as the ultraviolet LED at 3 W cm$^{-2}$, the reaction rate and quantum yield were smaller (blue triangles in Fig. 2d). Nevertheless, the reaction rate under lower energy photons exhibits an even higher exponent in the super-linear region ($n \approx 3.7$ at 523 K)[30]. At the highest intensity of the blue LED (4.9 W cm$^{-2}$), the EF and quantum yield at 623 K reached 27.8 ± 1.4 and 7.50%, respectively (blue diamonds in Fig. 2d). Unlike the sub-linear rate increase with increased light intensity characteristic of conventional semiconductor photocatalysis[12,13], this super-linear dependence indicates that very high reaction rates will not require very high light intensities.

To understand the mechanism, the reaction kinetics of $CO_2$ hydrogenation on Rh and Au photocatalysts in light and dark $H_2$-rich conditions were studied experimentally in the temperature range of 523 K and 623 K. The light intensity was chosen to be within the linear region to eliminate the effect of multiple excitation events. By fitting the measured temperature-dependent reaction rates with an Arrhenius equation, the apparent activation energy ($E_a$) of the thermo- and photo-reactions was obtained (Fig. 3). In virtually every case, the equation fits the data well, and the $E_a$ was ascertained with less than 5% uncertainty. For the thermocatalytic reactions on Rh, the $E_a$ for $CH_4$ and CO production was measured to be 78.6 ± 2 and 64.7 ± 6 kJ mol$^{-1}$

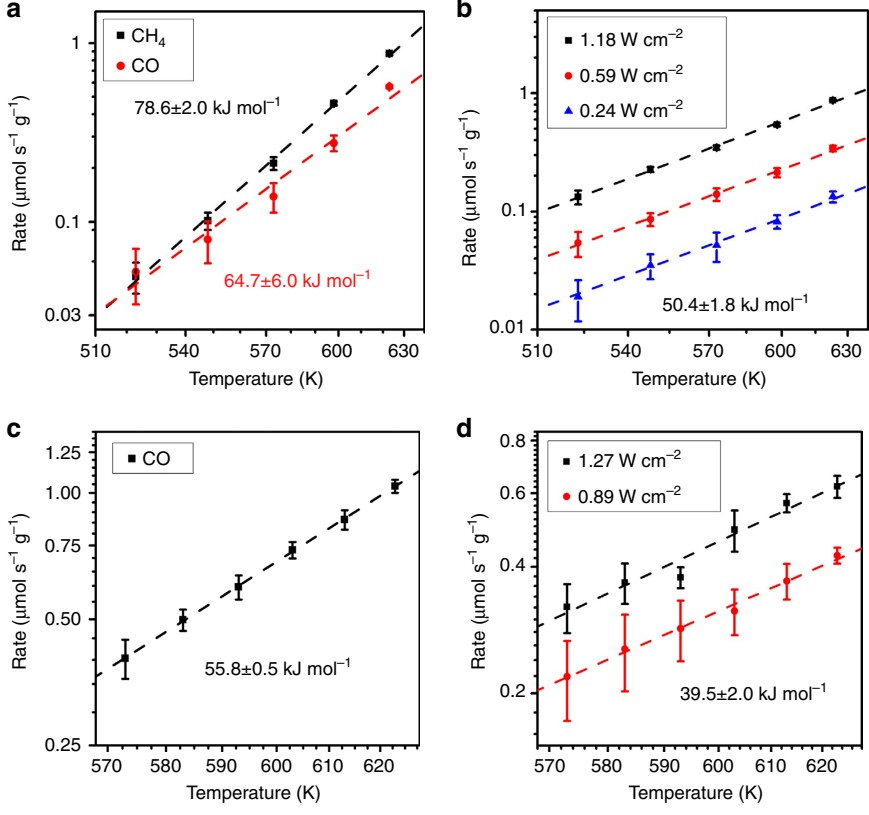

**Figure 3 | Apparent activation energies on the rhodium and gold photocatalysts.** (**a**) Thermocatalytic reaction rates of $CH_4$ (black squares) and CO (red circles) production on Rh/Al$_2$O$_3$ as a function of temperature. The apparent activation energies are obtained by fitting the results with an Arrhenius equation. (**b**) Photoreaction rates for $CH_4$ production on Rh/Al$_2$O$_3$ under 1.18 (black squares), 0.59 (red circles) and 0.24 W cm$^{-2}$ (blue triangles) ultraviolet illumination as a function of temperature. The photocatalytic reactions show the same apparent activation energy, which is lower than that of thermocatalytic reaction. (**c**) Thermocatalytic reaction rates of CO production on Au/Al$_2$O$_3$ as a function of temperature. (**d**) Photoreaction rates of CO production on Au/Al$_2$O$_3$ under 1.27 (black squares) and 0.89 W cm$^{-2}$ (red circles) white light as a function of temperature. Reduced apparent activation energies of photoreactions are observed on both Rh and Au photocatalysts, but with different selectivity. Error bars represent the s.d. of measurements by the mass spectrometer.

(0.81 eV and 0.67 eV), respectively, consistent with previous reports on supported Rh catalysts[51,64]. Under ultraviolet illumination, the photocatalytic $CH_4$ production showed a reduced $E_a = 50.4 \pm 1.8$ kJ mol$^{-1}$ (0.52 eV) for all light intensities. This reduction of $E_a$ was also observed for CO production on the Au photocatalyst with visible light, decreasing from $55.8 \pm 0.5$ kJ mol$^{-1}$ (0.58 eV) for the thermocatalytic reaction to $39.5 \pm 2$ kJ mol$^{-1}$ (0.41 eV) for photoreactions. The photoreaction rates of CO production on the Rh photocatalyst were too small for the activation energy to be deduced reliably.

The observed kinetics and selectivity of $CO_2$ hydrogenation on the Rh and Au photocatalysts shed light on the reaction mechanism of plasmonic photocatalysis. In thermocatalytic reactions, interactions between surface intermediates and catalysts dictate the propensity of competing pathways. For example, the higher selectivity towards $CH_4$ in the thermocatalytic reactions on Rh has already been ascribed to its higher $E_{ads,O}$ (ref. 61), while the exclusive selectivity for CO on the Au photocatalyst was caused by the low $E_{ads,O}$ and the deficiency of surface H atoms under our experimental conditions[65]. By contrast, in photoreactions, the transfer of hot electrons from plasmonic metal nanoparticles to specific intermediates critically depends on the energies of the hot electrons and the anti-bonding orbitals, thereby selectively activating certain reaction pathways and offering an additional means to tune the selectivity[23,24,37].

DFT calculations were carried out to understand how hot electrons affect the intermediates in the RDSs of $CH_4$ and CO production and explain the photo-selectivity we observed. The projected local density-of-states (LDOS) for adsorbed CHO and CO, key intermediates for $CH_4$ and CO production[47,51–58], respectively, are presented in Fig. 4a,b for the dominant Rh nanocube facet, Rh(100) (see 'Methods' section for details and Supplementary Fig. 7 for the configurations used in calculations). For clarity, only the orbitals involved in C–O bond cleavage for the Rh–CHO system and Rh–C bond cleavage for the Rh–CO system are plotted. The bonding interactions in both the CHO and CO systems lie ~6 eV below the Fermi level, suggesting a minimal role of hot holes in this process. For CHO, the C–O π* anti-bonding bands, which can accept hot electrons to weaken the C–O bond and facilitate $CH_4$ production, can be easily identified on the C($p_z$) and O($p_z$) orbitals at ~2 eV (Fig. 4a). On the other hand, the very weak and broad anti-bonding Rh-C interactions

observed on the C($p_x$) orbital at ~1 eV for CO (Fig. 4b) suggests a much smaller possibility of accepting ultraviolet photoexcited hot electrons by the CO intermediate compared with the CHO intermediate. Thus, the photo-generated hot electrons preferentially activated the CHO intermediate and enhanced $CH_4$ production, while only a small enhancement was observed for CO production. This mechanism is further verified by the lower selectivity towards $CH_4$ observed under non-resonant, lower energy blue light (~85%): the lower energy hot electrons had a lower probability of transferring to the higher energy anti-bonding orbital of the CHO intermediate (~2 eV) and a higher probability of transferring to the lower energy orbital of the CO intermediate (~1 eV). We note that due to the rapid decay via electron-electron and electron-phonon scatterings, the actual energies of the hot electrons are distributed below the associated photon energies of ultraviolet and blue light. Nevertheless, our computed relative magnitude of the LDOS peaks and the energy ordering for the relevant anti-bonding bands still offer a valid qualitative interpretation both for overall preference for $CH_4$ (under either ultraviolet or blue light) and for the slightly reduced $CH_4$ selectivity under blue light. A recent study using alloys similarly demonstrated the selective activation of certain reactants with different photon energies[26]. Generally speaking, the activation of a specific reaction intermediate using the absorption of specific photon energies by specific plasmonic metal nanostructures can specify product selectivity among competing reaction pathways.

The deduced process of thermo- and photocatalytic $CO_2$ hydrogenation on plasmonic Rh photocatalysts is summarized in Fig. 5. In the thermocatalytic reactions, phonons activate both CHO and CO intermediates and produce $CH_4$ and CO at comparable rates on the ground-state reaction coordinate (black curve in the bottom part of Fig. 5). In the photocatalytic reactions, hot electrons selectively transfer to the anti-bonding orbitals of CHO intermediates to weaken the chemical bonds and drive the reaction on a charged-state reaction coordinate characterized by a reduced activation energy (red curve in the top part of Fig. 5). This scenario is consistent with similar schemes proposed for other reactions on plasmonic metal photocatalysts[14–16,25,27]. In the future, red-shifting the plasmonic resonance of Rh nano-particles further into visible region, assembling Rh nanoparticles into closely packed clusters to create 'hot spots', and optimizing the reactant composition[47] could achieve even more selective and

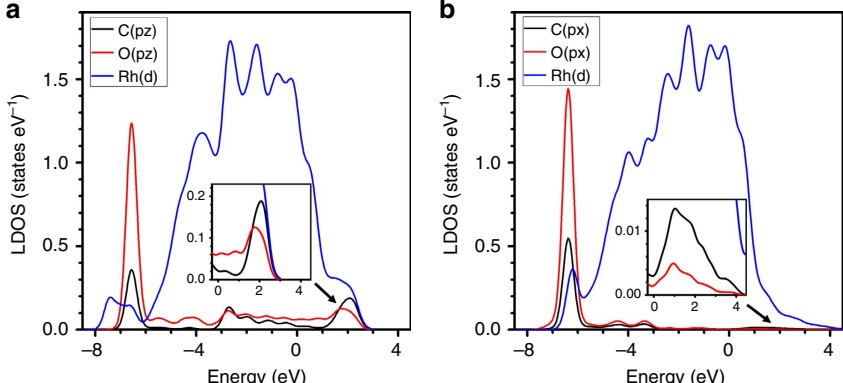

**Figure 4 | DFT calculations of CHO and CO intermediates on the Rh(100) surface. (a)** LDOS for adsorbed CHO on C($p_z$), O($p_z$), and Rh($d$) orbitals. The Rh(100) surface is perpendicular to the $x$ direction, and the C–O bond is along the $y$ direction. Major bands are identified as: (1) C–O π bonding band ($-6.5$ eV) with C($p_z$) (black) and O($p_z$) (red) interactions; (2) C–O π* anti-bonding band (1-3 eV, mode around 2 eV) with C($p_z$) and O($p_z$) interactions. **(b)** LDOS for adsorbed CO on C($p_x$), O($p_x$), and Rh($d$) orbitals. The Rh(100) surface is perpendicular to the $x$ direction, and the C–O bond is along the $x$ direction. Major bands are identified as: (1) C–O σ bonding band ($-6.3$ eV) with C($p_x$) (black) and O($p_x$) (red) interactions; (2) Very weak Rh-C anti-bonding band (0-3 eV, mode around 1 eV) with C($p_x$) and Rh($d$) (blue) interactions. The structures of the models used for calculations are given in Supplementary Fig. 7. All energies are referenced to the Fermi level. The insets are magnified plots of the anti-bonding regions.

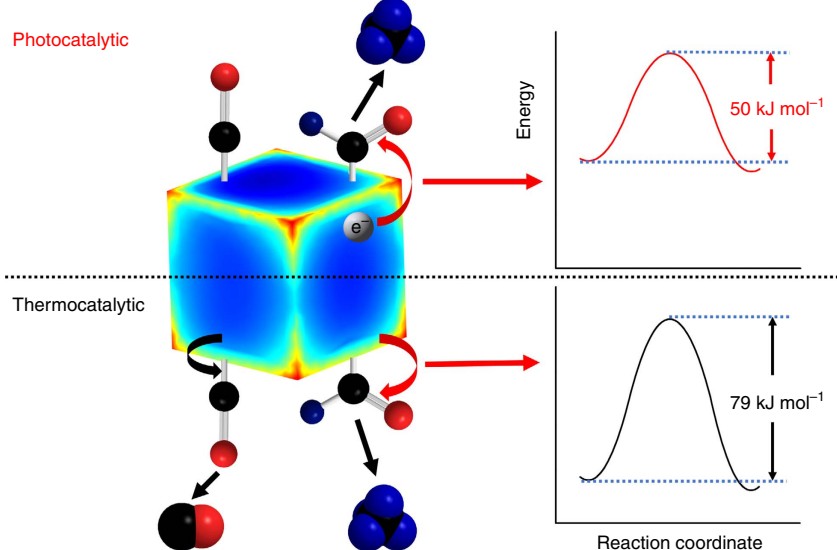

**Figure 5 | Reaction mechanism on a rhodium nanocube.** The thermocatalytic reaction activates both CO–Rh bonds and CH–O bonds to produce CO and CH$_4$, respectively. Hot electrons generated in the photocatalytic reaction selectively activate the C–O bonds of the CHO intermediate and reduce the apparent activation energy to enhance the CH$_4$ production rate. The black, red, and blue spheres are carbon, oxygen, and hydrogen atoms, respectively. The red corners of the cube show the intense electric field from the excitation of LSPRs[38].

efficient photocatalytic CH$_4$ production from CO$_2$ hydrogenation, even under direct or mildly concentrated sunlight. Our findings demonstrate that efficient plasmonic photocatalysis requires metals with both excellent catalytic and plasmonic properties. Although the mechanism analysed is based on CO$_2$ hydrogenation, the concept of selective activation of specific reaction intermediates to control the product selectivity can be applied to other plasmonic photocatalytic systems in ways that could prove to be transformative.

## Methods

**Photocatalyst preparation.** *Rh/Al$_2$O$_3$ photocatalyst.* The Rh nanocubes were synthesized by a modified slow-injection polyol method[38]. Overall, 54 mg potassium bromide (KBr, ACS, Acros) was dissolved in 2 ml ethylene glycol (EG, J. T. Baker) in a 20 ml glass vial and stirred in an oil bath at 160 °C for 1 h. 12 mg rhodium(III) chloride hydrate (RhCl$_3 \cdot x$H$_2$O, 38% Rh, Acros) and 25 mg polyvinylpyrrolidone (PVP, M.W. ≈ 55,000, Aldrich) were dissolved in 2 ml EG separately and injected into the hot reaction mixture by a two-channel syringe pump at a rate of 1 ml h$^{-1}$. The injection was paused for 15 min after adding 20 μl of the Rh precursor. After complete injection of the precursor, the reaction mixture was stirred for another 30 min and then cooled to room temperature. The suspension was washed with deionized water/acetone until no Cl$^-$ and Br$^-$ was detected in the supernatant. The solid was dispersed in 20 ml ethanol and impregnated on 90 mg Al$_2$O$_3$ nanoparticles (Degussa, Alu Oxide C, specific surface area 85–115 m$^2$ g$^{-1}$). The obtained solid was ground into powder and calcined in air at 400 °C for 2 h. The Rh nanocubes were well dispersed on the Al$_2$O$_3$ support and behaved as isolated nanoparticles.

*Au/Al$_2$O$_3$ photocatalyst.* A deposition-precipitation method was used to prepare highly dispersed small Au nanoparticles on support[15]. Overall, 100 mg Al$_2$O$_3$ nanoparticles were dispersed in 10 ml deionized water in a 20 mL glass vial by sonication. A total of 16 mg gold(III) chloride trihydrate (HAuCl$_4 \cdot x$H$_2$O, 99.9 + %, Aldrich) was added into the suspension and stirred in an oil bath at 80 °C. The pH was adjusted to ~8 by 1 M sodium hydroxide (NaOH) solution. After 4 h, the suspension was cooled and washed with copious deionized water/acetone until no Cl$^-$ was detected in the supernatant. The solid was dried at 100 °C overnight and calcined at 300 °C for 2 h.

**Reactor setup and photocatalytic reactions.** The photocatalytic reaction was carried out on a custom-built gaseous reaction system. Hydrogen (H$_2$, Research grade), carbon dioxide (CO$_2$, Research grade) and argon (Ar, UHP) were obtained from Airgas. The gas flow rates were controlled individually by mass flow controllers (Aalborg). Overall, ~15 mg of photocatalyst was loaded into the sample cup (diameter 6 mm, height 4 mm) in the reaction chamber (Harrick, HVC-MRA-5) for each experiment. The temperature was measured by a thermocouple

under the catalyst bed, and calculations indicate good thermal contact between the Rh nanoparticles and the surrounding media. The temperature of the photocatalyst bed was controlled by a PID temperature controller kit (Harrick, ATK-024-3) that managed the resistive heating power of the reaction chamber, and cooling water to mitigate heating caused by LED illumination. The photocatalysts were first reduced under 60.1 ml min$^{-1}$ H$_2$ and 27.6 ml min$^{-1}$ Ar at 350 °C for 4 h and then another 10.9 ml min$^{-1}$ CO$_2$ was introduced to achieve an H$_2$-rich CO$_2$:H$_2$ ratio of 1:5.5 and activate the photocatalysts for ~12 h to reach stable catalytic activities. The experiments with a H$_2$-deficient CO$_2$:H$_2$ ratio of 1:3.1 were conducted under 19.5 ml min$^{-1}$ CO$_2$, 60.1 ml min$^{-1}$ H$_2$ and 16.5 ml min$^{-1}$ Ar. Three LEDs with emission of 365 nm, 460 nm and 5700 K white light (Prizmatix, UHP-F) were used as light sources. The output power was controlled by a remote dial and measured with a thermopile power metre (Thorlabs, PM310D). The emission spectra of the light sources were measured with a CCD-based spectrometer (Thorlabs, CCS200). The gaseous product was analysed by a quadrupole mass spectrometer (Hiden, HPR-20) equipped with a Faraday cup detector. The detection limit of the mass spectrometer is ~0.001% conversion of CO$_2$. The reactions were all operated in the low-conversion and light-controlled regime. For each temperature and light intensity condition, at least 15 min elapsed before reaching steady state and seven sequential measurements were made to determine the steady-state concentration of each gas and the associated reaction rates and uncertainties. The 15 atomic mass unit (amu) signal was used to quantify the methane production rate. The 28 amu signal was used to quantify the carbon monoxide production rate, from which the background from carbon dioxide feedstock was subtracted. Deuterium (D$_2$, Sigma Aldrich, 99.8% atom D) was used in place of H$_2$ for the isotopic labelling experiments.

**DFT calculations.** All calculations in this work were performed with the Vienna Ab initio Simulation Package (VASP)[66]. The Perdew–Burke–Ernzerhof (PBE)[67] exchange-correlation functional was used along with its corresponding projected augmented wave (PAW) pseudopotentials. The semi-empirical D2 model[68] was applied to describe the Van der Waals interactions. A plane-wave cutoff of 500 eV was chosen. The Gamma centred $1 \times 2 \times 2$ k-point was used for the structural relaxations (converged to 0.01 eV Å$^{-1}$), and $1 \times 8 \times 8$ for the projected LDOS calculations. Periodic boundary conditions were used in all three directions for the face-centred cubic (*fcc*) Rh model (Supplementary Fig. 7). A vacuum of 15 Å was used in the *x* direction to separate the Rh(100) surface slabs containing four layers of Rh atoms. In the *y* and *z* directions the lattice size for the supercell was chosen to be three times that of a unit cell. The adsorbed CHO and CO groups were placed on the exposed Rh(100) surface.

**Material characterization.** Transmission electron microscopy (TEM) images were collected by a FEI Tecnai G$^2$ Twin operating at 200 kV. The TEM samples were prepared by dispersing the photocatalysts in ethanol with sonication and depositing on a copper grid coated with a carbon film (Ted Pella, 01813). Diffuse-reflectance ultraviolet–visible extinction spectra were obtained on an Agilent Cary 5,000 equipped with an external diffuse-reflectance accessory

(DRA-2500). The composition of the photocatalysts was measured by a Kratos Analytical Axis Ultra X-Ray Photoelectron Spectrometer.

**Data availability.** The data that support this study are available from the corresponding authors on request.

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

## Acknowledgements

This research is supported by the National Science Foundation (CHE-1565657) and the Army Research Office (Award W911NF-15-1-0320). X.Z. is supported by Katherine Goodman Stern fellowship from the Graduate School, Duke University. X.L. is supported by the Department of Defense (DoD) through the National Defense Science & Engineering Graduate Fellowship (NDSEG) Program. D.Z., N.Q.S. and W.Y. are supported by the Center for the Computational Design of Functional Layered Materials, an Energy Frontier Research Center funded by the U.S. Department of Energy (DOE), Office of Science, Basic Energy Sciences (BES), under Award # DE-SC0012575. We acknowledge helpful conversations about plasmonic photocatalysis with P. Christopher and N. Halas and thank A. Barreda, Y. Gutierrez, F. González and F. Moreno for the prior electromagnetic simulations, and M. Therien and B. Langloss for the help with diffuse-reflectance extinction spectroscopy. We also acknowledge the support from Duke University SMIF (Shared Materials Instrumentation Facilities).

## Author contributions

H.O.E. conceived the project with J.L., and X.Z., J.L. and H.O.E. devised and developed the experiment. X.Z. carried out experimental work and analysis, and X.L. contributed to the analysis of the data and proofread the manuscript. D.Z., N.Q.S. and W.Y. carried out the DFT calculations. All the authors wrote the manuscript. J.L. is the PhD advisor and H.O.E. is the co-advisor of X.Z. and X.L. W.Y. is the advisor of D.Z. and N.Q.S.

## Additional information

**Competing financial interests**: The authors declare no competing financial interests.

**How to cite this article**: Zhang, X. *et al.* Product selectivity in plasmonic photocatalysis for carbon dioxide hydrogenation. *Nat. Commun.* **8,** 14542 doi: 10.1038/ncomms14542 (2017).

**Publisher's note**: 

