## [Peer Review File · Nature Communications]

Reviewers' comments:

Reviewer #1 (Remarks to the Author):

This is a revised version. The authors have addressed a number of issues and refocused the paper on the modification of selectivity by light excitation. The DFT simulations also add some insights. There are major weaknesses that remain.

1) The authors continue to refer to the photocatalysis as plasmonic, although there is no conclusive evidence of this, an objection raised by another reviewer as well. Yes, the Rh nanoparticles have UV plasmon resonances. That is no under contention. But there is no way to separate the plasmonic absorption from the interband absorption of the metal, unless the plasmon resonance is tuned to the red region of the visible spectrum. Thus, in the present case, there is no way to selectively excite the plasmon resonance. In fact, UV or blue excitation invariably cause interband excitation, inducing the generation of e^-/h^+ pairs. The fact that blue light (off-resonant from the plasmonic peak) also leads to photocatalytic conversion is proof for the latter explanation.

2) Utility of the work for solar-to-fuel conversion is low, because the photocatalysis requires UV light. UV-induced photocatalytic CO_2 conversion or water splitting has not been a major objective/hurdle. It is the need to induce such processes with visible light that is a major pursuit.

3) The DOS calculations based on DFT need to be consistent in terms of their ability to explain the selectivities observed. It is not clear how the authors use the DOS results to explain that CH_4 and CO are produced at comparable rates in the absence of light. Likewise the explanation of selectivity under blue light sounds incorrect. Blue light is ca. 3 eV, so why is there preference for a level 1 eV higher than the Fermi level as compared to a level 2 eV above the Fermi level?

Reviewer #2 (Remarks to the Author):

The major points are the product selectivity for CH_4 especially in the photocatalytic reaction of CO_2 hydrogenation. Authors reported that mild illumination of Rh NPs was critical in the selectivity which was demonstrated with comparative experiment and DFT calculation. The finding is looking novel and very important in this area. Authors also well documented the experimental procedures and statistical analysis. Therefore I suggest the acceptance of submitted manuscript in current format but with some minor revision.

1. Authors claimed the mild illumination for the selectivity but it is hard to separate the photothermal effect and normal heating effect. As authors displayed temperature dependent CH_4 selectivity in Fig.2b, when illuminate catalyst with UV and blue LED, the localized thermal effect on Rh-NPs is unavoidable. The CH_4 selectivity in Fig. 2b in high temperature (500-600K) is also very high, which is contradictory results compare with the main finding of authors. This is not clear yet that need to clarify.

2. In Fig. 3, the calculated activation energy is good to support the experimental results. Therefore, I like to recommend the addition of detailed calculation in the supporting information.

We appreciate the helpful comments from the reviewers and revised the manuscript significantly to clarify our finding. The detailed responses and quotations from the revised manuscript are listed below.

Reviewer #1:

This is a revised version. The authors have addressed a number of issues and refocused the paper on the modification of selectivity by light excitation. The DFT simulations also add some insights. There are major weaknesses that remain.

1) The authors continue to refer to the photocatalysis as plasmonic, although there is no conclusive evidence of this, an objection raised by another reviewer as well. Yes, the Rh nanoparticles have UV plasmon resonances. That is no under contention. But there is no way to separate the plasmonic absorption from the interband absorption of the metal, unless the plasmon resonance is tuned to the red region of the visible spectrum. Thus, in the present case, there is no way to selectively excite the plasmon resonance. In fact, UV or blue excitation invariably cause interband excitation, inducing the generation of e⁻/h⁺ pairs. The fact that blue light (off-resonant from the plasmonic peak) also leads to photocatalytic conversion is proof for the latter explanation.

Response: We thank the reviewer for raising a question many readers might have regarding plasmonic vs interband absorption. The question is probably motivated by knowledge of the Au band structure, but the Rh band structure is very different and not nearly as well known. We have juxtaposed the two band structures here to explain the difference. *We will assume the reviewer and the readers will agree that a necessary but not sufficient condition for plasmonic behavior is that the metal exhibits nearly free electron behavior (i.e. quasi-quadratic band structure).*

In the Au band structure (Fig. 1a, modified from T. Rangel, *et al. Phys. Rev. B*, 2012, 86, 125125), it can be seen that free electron and therefore plasmonic behavior cannot extend above about 2.5 eV (red region in Fig. 1a) because of strong interband absorption, an effect well known in the plasmonic community that limits gold's plasmonic performance to longer wavelengths. *It is particularly important to note that this interband transition moves relatively free electrons near the Fermi energy to a flat band with low mobility.*

Similar free-to-flat interband transitions occur in Rh too (Fig. 1b, modified from N. E. Christensen, *phys. stat. sol. (b)*, 1973, 55, 117), but they stop at 1.8 eV (700 nm) and don't begin again until 8.6 eV (145 nm, red regions in Fig. 1b). The quasi-quadratic band structure in between indicates that absorbed red to deep UV photons drive comparatively free electrons that can manifest plasmonic behavior in Rh nanostructures, as the many cited prior calculations and measurements have already confirmed. The plasmonic resonance of the Rh photocatalyst powder and Rh cube suspensions used in our experiments occurs in the near UV (3.71 eV, 334 nm). Since the conductivity of Rh is lower than better plasmonic metals like Ag and Al, its resonance is broad and extends into the blue. *Thus the experiments*

performed with 3.4 and 2.7 eV (365 and 460 nm) photons overlap the broad plasmon resonance of Rh and not its flat interband transitions.

So our measurements occur in the same nearly free electron regime of Rh as prior plasmonic photocatalytic demonstrations using Au, Ag, Cu, and Al (refs. 14-22), by which plasmonic excitation quickly decayed through e-/h+ pair generation, some of which catalyze reactions on adsorbed species. We trust the reviewer will agree that the posed interband excitation problem in Rh is in the red, not the blue or UV, just like it is in Al where the plasmonic and interband photocatalytic effects were separately observed (ref. 16). In fact, plasmonic and unwanted interband excitations begin to overlap in the red for Rh, while the plasmonic photocatalytic behavior is best observed in the relatively free electron UV-blue region we operate.

We had hoped our copious references to prior work demonstrating the plasmonic behavior of Rh would be sufficient for this brief communication, and we were reluctant to rehearse the Rh band structure since that goes well beyond the scope of our manuscript. Nevertheless, since this question seems to be the principal concern about our manuscript, we briefly discussed the band structure of Rh in the manuscript and included as a supplement the Rh band diagram and a brief tutorial caption summarizing these points as a “for review only” document.

In addition, we have revised several sentences in the manuscript to make these points more clearly:

p. 3 “Recently, it has been discovered that plasmonic metal nanoparticles are photocatalytically active,¹⁴⁻²⁹ driving chemical reactions with photo-generated hot carriers and exhibiting a compelling super-linear dependence on light intensity ($R_{photo} \propto I^n, n > 1$)^{18,28,30}.”

p. 4 “The distribution of photo-excited carriers depends on the local density of states in the metal and the associated band structure, the LSPR of the nanostructure, and the energy of the photon³⁵.”

p. 4 “Recently, the size- and shape-dependent plasmonic properties of rhodium (Rh) nanoparticles have been demonstrated at energies tunable throughout the UV and visible regions³⁸⁻⁴³. Like Au and Pt, Rh is a transition metal without a native oxide coating, and direct bonding between adsorbates and the metal surface greatly facilitates the transfer of hot carriers for plasmonic photocatalysis.”

p. 6 “The band structure of Rh⁴⁹ indicates that the UV and blue excitations avoid lower energy parasitic interband absorption and generate nearly free hot electrons with energies 2.5 and 2.1 eV above the Fermi level^{35,41}, respectively.”

2) Utility of the work for solar-to-fuel conversion is low, because the photocatalysis requires UV light. UV-induced photocatalytic CO₂ conversion or water splitting has not been a major objective/hurdle. It is the need to induce such processes with visible light that is a major pursuit.

Response: We certainly agree with the reviewer that there is little UV light from sunlight and that there is a desire to demonstrate CO₂ conversion with sunlight. ***However, this is never claimed to be the “main pursuit” of this manuscript.*** The only time we mention this possibility is in our concluding paragraph to foreshadow what can be done next. As important as solar CO₂ conversion is to the photocatalysis community, the selective production of a target product among a spectrum of possible hydrocarbons, alcohols, and oxides could be even more transformative, and that is the main point of this manuscript. To be clear, the important discovery being reported here is the plasmonic photocatalytic activity of Rh nanostructures and their selective activation of a specific reaction pathway among competing pathways by injecting plasmon-generated hot electrons into specific reaction intermediate.

(We should note that our foreshadowing is done confidently: we have already demonstrated that larger Rh nanostructures have LSPRs in the visible wavelength region (ref. 38), and interband absorption may play a more important role when much larger Rh nanostructures are used.)

To avoid any misleading language about our primary claims, we have modified the manuscript accordingly:

p. 14 “In the future, red-shifting the plasmonic resonance of Rh nanoparticles farther into visible region, assembling Rh nanoparticles into closely packed clusters to create “hot spots”, and optimizing the reactant composition⁴⁷ could achieve even more selective and efficient photocatalytic CH₄ production from CO₂ hydrogenation, even under direct or mildly concentrated sunlight.”

3) The DOS calculations based on DFT need to be consistent in terms of their ability to explain the selectivities observed. It is not clear how the authors use the DOS results to explain that CH₄ and CO are produced at comparable rates in the absence of light. Likewise the explanation of selectivity under blue light sounds incorrect. Blue light is ca. 3 eV, so why is there preference for a level 1 eV higher than the Fermi level as compared to a level 2 eV above the Fermi level?

Response: We appreciate the reviewer’s insightful and constructive comments on the DFT calculations and the product selectivity. We have clarified the manuscript to address these excellent questions.

First of all, we are not attempting to “use the DOS results to explain that CH₄ and CO are produced at comparable rates in the absence of light.” Our calculated DOS results only serve to explain the selectivity under the photocatalytic conditions where the hot electron injection into the antibonding bands is involved. The manuscript indicates that the selectivity and production rates in the absence of light is already understood (see references 47 and 51-58), and it would require a different computational approach involving the adsorption free energies of reaction intermediates based on Sabatier’s principle (*J. Am. Chem. Soc.* **127**, 5308 (2005)).

So let us consider the comment “Blue light is ca. 3 eV, so why is there preference for a level 1 eV higher than the Fermi level as compared to a level 2 eV above the Fermi level?” We are not claiming that there is a preference for the 1 eV level (CO production) as compared to the 2 eV level (CH₄ production), only that there is a slightly reduced preference for CH₄ production with blue light (90% for 2.7 eV photons) than with UV light (95% for 3.4 eV photons). This is easily understood as a consequence of the higher starting energy for the hot electrons produced by UV light. Plasmonically photogenerated hot electrons rapidly decay via electron-electron and electron-phonon scatterings, forming a distribution of energies that broadens and lowers with time. Thus, for a given time after photoexcitation, a larger portion

of UV-generated hot electrons have enough energy (~ 2 eV above E_f) to transfer into the higher-energy C-O anti-bonding orbitals of the CHO intermediate than do the hot electrons generated by lower energy blue photons. The hot electrons that do not have enough energy to transfer to the CHO intermediate may still transfer to the lower energy anti-bonding Rh-C orbital of the CO intermediate (~ 1 eV above E_f). Because the distribution of blue-generated hot electrons is lower than for the UV-generated hot electrons, the selectivity towards CH_4 under blue light is slightly lower than under UV illumination, but it is still much higher than the selectivity towards CO. In summary, our calculated energy ordering for the relevant antibonding orbitals of the CHO and CO intermediates is consistent with the slightly reduced selectivity towards CH_4 under blue light, and our calculated relative LDOS magnitudes agree with the overall preference for CH_4 under both UV and blue light.

With this regard, we added the following sentence:

p.13: “We note that due to the rapid decay via electron-electron and electron-phonon scatterings, the actual energies of the hot electrons are distributed below the associated photon energies of UV and blue light. Nevertheless, our computed relative magnitude of the LDOS peaks and the energy ordering for the relevant antibonding bands still offer a valid qualitative interpretation both for overall preference for CH_4 (under either UV or blue light) and for the slightly reduced CH_4 selectivity under blue light.”

Reviewer #2:

The major points are the product selectivity for CH₄ especially in the photocatalytic reaction of CO₂ hydrogenation. Authors reported that mild illumination of Rh NPs was critical in the selectivity which was demonstrated with comparative experiment and DFT calculation. The finding is looking novel and very important in this area. Authors also well documented the experimental procedures and statistical analysis. Therefore I suggest the acceptance of submitted manuscript in current format but with some minor revision.

We appreciate your positive review and recommendation that our manuscript be accepted! The answers to the questions you raised are detailed here.

1. Authors claimed the mild illumination for the selectivity but it is hard to separate the photothermal effect and normal heating effect. As authors displayed temperature dependent CH₄ selectivity in Fig.2b, when illuminate catalyst with UV and blue LED, the localized thermal effect on Rh-NPs is unavoidable. The CH₄ selectivity in Fig. 2b in high temperature (500-600K) is also very high, which is contradictory results compare with the main finding of authors. This is not clear yet that need to clarify.

Response:

We thank the reviewer for this comment, because we thought we had adequately addressed this in response to another reviewer's comment, but apparently we hadn't! Certainly our LEDs increase the temperature of the Rh photocatalyst, as demonstrated by a control experiment in which the temperature of the uncooled photocatalyst bed increased by ~25 K under 3.0 Wcm⁻² UV illumination. Indeed, our experiments required a combination of cooling water and resistive heating managed by a temperature controller to keep the photocatalyst bed at the desired temperature (400-650K). Only at room temperature did we turn off the temperature control, and the resulting 29K temperature rise was caused by a combination on LED heating (25K) and reaction exothermicity (4K). All this was indicated on pages 6 and 10 of our submitted manuscript.

To make this even clearer, we expanded a sentence in the Methods section:

p. 16 "The temperature was measured by a thermocouple under the catalyst bed, and calculations indicate good thermal contact between the Rh nanoparticles and the surrounding media. The temperature of the photocatalyst bed was controlled by a PID temperature controller kit (Harrick, ATK-024-3), managing the resistive heating power of the reaction chamber, and cooling water to mitigate heating caused by LED illumination."

We fear we may not have understood the rest of the question being asked, but we think the reviewer is suggesting that the high selectivity we see at high temperatures somehow contradicts our findings; specifically, that selectivity should decrease with increasing temperature as thermal non-selectivity supersedes plasmonic selectivity. This is an insightful observation, and we appreciate the opportunity to address it. Indeed, the high selectivity

towards CH₄ at high temperature shows another advantage of plasmonic photocatalysis compared to conventional photocatalysis, where reaction rate could decrease with increasing reaction temperature. Illumination works selectively only on the critical intermediate. The energies involved with selecting that intermediate are not accessible solely by thermal excitation, but high temperature can significantly increase the reaction rate in plasmonic photocatalysis. Therefore, increasing the temperature does not affect the selectivity, but the combination of light and heat produces selectivity and rates not possible with either alone.

Although we indicated this on p. 9, we changed revised manuscript to clarify this point further:

p. 9 “The rather modest local heating in our experiments and the observed high photo-selectivity towards CH₄ over CO indicate that the photo-enhanced reaction rates do not originate from thermal or plasmonic photothermal heating on the Rh nanoparticle surface.”

p. 9 “Although the selectivity is derived almost entirely from light, heat significantly increases the reaction rate.”

2. In Fig. 3, the calculated activation energy is good to support the experimental results. Therefore, I like to recommend the addition of detailed calculation in the supporting information.

Response:

We fear our plot may have misled the reviewer. The activation energies (E_a) represented in Fig. 3 are not calculated but fitted from the experimentally measured rates (r) using the Arrhenius equation

$$r = a \times e^{\frac{-E_a}{RT}}.$$

The activation energies we measured for the thermocatalytic reactions on the Rh catalyst are in good agreement with previously reported values (refs 51, 64). Our measurements show that a single activation energy (50.4 kJ/mol) for the plasmonic photocatalytic CH₄ reaction is both able to reproduce the measured rates as a function of temperature and prove that heating does not affect selectivity. There is no more detailed calculation to report.

To clarify that our activation energies are measured and fitted, not calculated ab initio, we modified manuscript to clarify this point:

p. 11 “To understand the mechanism, the reaction kinetics of CO₂ hydrogenation on Rh and Au photocatalysts in light and dark conditions were studied **experimentally** in the temperature range of 523 and 623 K.”

p. 11 “By fitting the **measured** temperature-dependent reaction rates with an Arrhenius equation, the apparent activation energy (E_a) of the thermo- and photo-reactions was obtained (Fig. 3).”

Regarding the request for a “detailed calculation”, we fear we may have again misunderstood the reviewer. We refer the reviewer to our response to question 3 by the other reviewer in hopes that answers your suggestion too. If it doesn’t and we are being asked to calculate this activation energy *ab initio*, we agree it would be helpful, but we would prefer to feature this rather complex calculation in a subsequent publication instead of having it lost in a supplement. This has been recently calculated in the Ru system for this same reaction (Ref. 47), which is similar to the Rh system we studied here for many fundamental reasons. Anyway, we are convinced that our data provides the most compelling proof of our assertions. The DFT calculations presented here were designed to help us understand the observed selectivity, which was the primary objective of our “communication.”

REVIEWERS' COMMENTS:

Reviewer #1 (Remarks to the Author):

Thank you for your responses and revisions, which are satisfactory. I suggest that the band structure of Rh, included as supporting information for review purposes, become a part of the supporting information for publication.

Reviewer #2 (Remarks to the Author):

At this moment, I think the revised manuscript contain a considerate discussions about the results they obtained. The result and discussion is found to be sound and acceptable. I like to recommend the acceptance of this manuscript in Nature Communications. It is interesting for me to review this manuscript.

Reviewer #1:

Thank you for your responses and revisions, which are satisfactory. I suggest that the band structure of Rh, included as supporting information for review purposes, become a part of the supporting information for publication.

Response: We thank the reviewer for the helpful suggestions. We included the band structure of Rh as Supplementary Fig. 2 in the supplementary information and obtained the copyright permission from the publisher of the reference. The readers now have the access to the band structure of Rh to understand that the light used in this manuscript excite the plasmonic behaviors of Rh.

Reviewer #2:

At this moment, I think the revised manuscript contain a considerate discussions about the results they obtained. The result and discussion is found to be sound and acceptable. I like to recommend the acceptance of this manuscript in Nature Communications. It is interesting for me to review this manuscript.

Response: We also appreciate your recommendation of acceptance and helpful comments in the reviewing process and are pleased to have our manuscript be reviewed by you.